# Secondary Metabolites Produced by the Blue-Cheese Ripening Mold *Penicillium roqueforti*; Biosynthesis and Regulation Mechanisms

**DOI:** 10.3390/jof9040459

**Published:** 2023-04-10

**Authors:** Renato Chávez, Inmaculada Vaca, Carlos García-Estrada

**Affiliations:** 1Departamento de Biología, Facultad de Química y Biología, Universidad de Santiago de Chile (USACH), Santiago 9170022, Chile; 2Departamento de Química, Facultad de Ciencias, Universidad de Chile, Santiago 7800003, Chile; 3Departamento de Ciencias Biomédicas, Facultad de Veterinaria, Campus de Vegazana, Universidad de León, 24071 León, Spain

**Keywords:** *Penicillium roqueforti*, blue cheese, secondary metabolism, andrastin, roquefortine, PR-toxin, mycophenolic acid, festuclavine, isofumigaclavine, annullins

## Abstract

Filamentous fungi are an important source of natural products. The mold *Penicillium roqueforti*, which is well-known for being responsible for the characteristic texture, blue-green spots, and aroma of the so-called blue-veined cheeses (French Bleu, Roquefort, Gorgonzola, Stilton, Cabrales, and Valdeón, among others), is able to synthesize different secondary metabolites, including andrastins and mycophenolic acid, as well as several mycotoxins, such as Roquefortines C and D, PR-toxin and eremofortins, Isofumigaclavines A and B, festuclavine, and Annullatins D and F. This review provides a detailed description of the biosynthetic gene clusters and pathways of the main secondary metabolites produced by *P. roqueforti*, as well as an overview of the regulatory mechanisms controlling secondary metabolism in this filamentous fungus.

## 1. Introduction

Filamentous fungi are prolific producers of secondary metabolites. These metabolites are a wide group of diverse organic molecules that contribute to fundamental biological processes in fungi, including defense, communication with other microorganisms, and virulence in pathogenic interactions [1,2]. Many fungal secondary metabolites are bioactive molecules with medical and/or commercial interests [1,3]. Therefore, their study is an exciting topic of scientific and applied interest.

*Penicillium* is one of the most important fungal genera in the field of secondary metabolites. The most emblematic example is *Penicillium chrysogenum* (reclassified as *Penicillium rubens*), the industrial producer of penicillin [4,5]. Beyond *P. chrysogenum*, several studies have highlighted the potential of other members of the genus *Penicillium* as potential producers of a vast and diverse array of secondary metabolites of interest [6,7,8,9]. Among these fungi, *Penicillium roqueforti* emerges as one of the most interesting species. During the last years, much progress has been made in the secondary metabolism of this fungus. In this review, we summarize and update the main aspects of the secondary metabolism of *P. roqueforti* focusing on the metabolites produced by this fungus, their biosynthesis, and regulatory mechanisms.

## 2. Brief Overview of Taxonomic and Biotechnological Aspects of *P. roqueforti*

*P. roqueforti* is a saprophytic filamentous fungus (mold) whose colonies show a color from light to dark greenish gray and can include gray, yellowish, and olive-green shades. It also has a texture that can vary from velvety to fasciculate. Conidiophores constitute a velutinous felt with phialides, which produce spherical, smooth, and dark green conidia (3 to 4.5 μm diameter) included in terminal penicilli, which are typically terverticillate (quaterverticillate and more rarely biverticillate can also be observed) [10].

The original taxonomic description of the species *P. roqueforti* was performed by Thom in 1906 [11], using a strain isolated from a Roquefort cheese purchased in a market in the United States. Despite this early taxonomic description, the taxonomy of this species has been complex. In the past, the denomination *P. roqueforti* included a group of heterogeneous fungi (the “*P. roqueforti* group”) with very similar phenotypic characteristics, which are hard to distinguish by the traditional morphological and physiological methods [12]. In addition, through the years, numerous strains of *P. roqueforti* independently isolated by different researchers were designated with different names, making more complex the establishment of accurate taxonomic denominations for new isolates. With the advent of molecular techniques, the taxonomy of the “*P. roqueforti* group” began to be clarified [12]. In 2004, Frisvad and Samson accomplished the full taxonomy of this species synonymizing many of the different names given to *P. roqueforti* over the years and designating *P. roqueforti* IMI 024313 as the neotype for the species [13].

Currently, *P. roqueforti* is classified within the subgenus *Penicillium*, section *Roquefortorum*, and series *Roquefortorum* along the closely related species *P. carneum*, *P. mediterraneum*, *P. paneum,* and *P. psychrosexuale* [14]. As all the members of this series, *P. roqueforti* is characterized by having large globose conidia and rough-walled conidiophore stipes, the ability to grow at elevated carbon dioxide levels, and the ability to produce the mycotoxin roquefortine C, but it can be clearly distinguished from the other members of the series by phylogenetic analyses using concatenated markers, its ability to produce specific secondary metabolites, and its capability of performing heterothallic sexual reproduction [14,15].

*P. roqueforti* has been used for centuries as a maturation agent in blue cheese, which includes several varieties such as French Bleu and Roquefort, Italian Gorgonzola, English Stilton, Spanish Cabrales, Picón Bejes-Tresviso and Valdeón, and many others from Denmark and the United States [10,16]. Blue cheese receives this name due to the blue-veined appearance that results from the proliferation of the melanized fungal conidia (asexual spores) within aerated cavities in the cheese [17]. *P. roqueforti* spores spontaneously contaminating milk were the origin of blue cheese. However, since the end of the 18th century, conidia are inoculated during the production process [18,19], which is slightly different according to the variety. This process mainly comprises the inoculation of the ripening mold *P. roqueforti* in liquid suspensions, which are added either to milk batches containing high levels of fat obtained from sheep or cow, depending on the variety, or the curd. Fermentation of this type of cheese is carried out by mesophilic lactic acid bacteria: *Streptococcus lactis*, *Streptococcus lactis* subsp. *diacetylactis* and *Leuconostoc* spp. [16]. In addition, *P. roqueforti* acts as a secondary starter providing cheese with a characteristic intense and spicy flavour because of the proteolytic and lipolytic activities, which, during ripening, generate volatile and non-volatile compounds responsible for the mouldy aromas [20,21]. During the maturation process, other microorganisms, such as *Brevibacterium linens*, also proliferate and add specific aroma to many blue cheeses [22].

In addition to the main use of this microorganism in the production of blue cheese, *P. roqueforti* has also been considered for other biotechnological purposes, such as the production of different metabolites, including the immunosuppressant agent mycophenolic acid [23], lipase extracts on solid state fermentation using cocoa shells as a substrate [24], or cellulolytic enzyme extracts upon cultivation on residue of yellow mombin fruit [25].

## 3. Secondary Metabolites Produced by *P. roqueforti*

For many years, *P. roqueforti* has been known to produce several secondary metabolites with biological properties, which encouraged the chemical study of this fungal species. As a result, numerous secondary metabolites were purified for the first time from *P. roqueforti* cultures, and their structures were clarified. In addition, other compounds already known to be produced by other fungi are also produced by *P. roqueforti*. A brief overview of these secondary metabolites (Figure 1) is described below.

### 3.1. PR-Toxin and Related Compounds

During the 70s, a potent toxin from *P. roqueforti*, named PR-toxin (Figure 1a), was discovered by Wei et al. [26]. The compound was purified from stationary cultures of the fungus on liquid YES medium (2% yeast extract, 15% sucrose), and its toxicity to rats was demonstrated [26]. In this original description, the chemical structure of PR-toxin was only partially elucidated. The full elucidation of its structure was achieved in a subsequent work [27]. From a chemical point of view, PR-toxin belongs to the eremophilane terpenoid class, it is a bicyclic sesquiterpene with the presence of two stable epoxide rings, and it has several functional groups [28].

PR-toxin is unstable and can be readily converted into other compounds. This property led to the identification of some PR-toxin related metabolites. Chang et al. [29] found that two compounds appeared in the culture medium of *P. roqueforti,* while PR toxin decreased. These compounds were purified, and their structures were elucidated, revealing that they were PR-imine and PR-amide (also known as Eremofortin E) [29]. Later, the same group purified and identified a third degradation product, which was named PR-acid [30]. 

As was mentioned before, PR-toxin belongs to the eremophilane terpenoid class. Accordingly, eremophilane compounds related to PR-toxin have also been purified and characterized. In this way, in three consecutive papers, Moreau and co-workers reported five eremofortins (A–E) related to PR-toxin, which were isolated from filtrates of *P. roqueforti* [31,32,33].

### 3.2. Roquefortine C and Related Compounds

Roquefortines are a family of prenylated diketopiperazine indole alkaloids, whose core structure is formed by the condensation of L-tryptophan and L-histidine [34]. Roquefortines are produced by several fungi from the genus *Penicillium* [6]. The main member of this family is Roquefortine C (Figure 1b), which is also one of the major secondary metabolites produced by *P. roqueforti* [34]. Roquefortine C was originally isolated from cultures of *P. roqueforti* in 1975 [35], although its full structural elucidation was achieved a couple of years after [36,37].

In addition to roquefortine C, the other member of the family that has been isolated from cultures of *Penicillium roqueforti* is Roquefortine D [38]. At this point, it should be mentioned that there are other members of the roquefortine family as well as related compounds (meleagrin, glandicolins), but they are not produced by *P. roqueforti* [34]. This point will be revisited below.

### 3.3. Other Secondary Metabolites Produced by P. roqueforti

PR-toxin and Roquefortine C are the main toxic secondary metabolites produced by *P. roqueforti*, so they were treated separately before. However, this fungus can produce other minor mycotoxins as well as non-toxic compounds, which are briefly described in this section.

In early studies on purification of Roquefortine C, some ergot alkaloids were co-purified [35]. They included festuclavine and two other compounds that were originally named Roquefortine A and Roquefortine B, which correspond to the ergot alkaloids Isofumigaclavine A (Figure 1c) and Isofumigaclavine B, respectively [35]. In addition, another ergot alkaloid, agroclavine, was detected by mass spectrometry in *P. roqueforti* [39].

Another interesting compound produced by *P. roqueforti* is mycophenolic acid (Figure 1d). This compound is a meroterpenoid with a phthalide moiety that exhibits several biological activities, being widely used as immunosuppressant for the prevention of organ transplant rejection [40]. Mycophenolic acid is commonly detected in many *P. roqueforti* strains [39,41].

Andrastins A–D are a family of meroterpenoid compounds derived from dimethyl orsellinic acid, which are interesting candidates as anticancer drugs [34]. Andrastins, especially the main member of the family, Andrastin A (Figure 1e), are usually detected in *P. roqueforti* [39,41,42].

In recent years, a family of sesterterpenoid compounds were isolated from *P. roqueforti*. The first compounds isolated from this family were the pentacyclic sesterterpenes Peniroquesines A–C [43]. Later, Roquefornine A (Figure 1f), another sesterterpene with an unprecedented pentacyclic system and cytotoxic activity, was described by Wang et al. [44]. Finally, seven other compounds related to Peniroquesines A–C were recently described, some of them showing interesting cytotoxic and anti-inflammatory activities [45]. 

## 4. Biosynthetic Gene Clusters and Pathways Functionally Characterized in *P. roqueforti*

During the last decade, several research groups were able to elucidate the gene clusters and enzymatic pathways responsible for the biosynthesis of the main secondary metabolites produced by *P. roqueforti*, i.e., Roquefortine C, Isofumigaclavine A, PR-toxin, Andrastin A, and mycophenolic acid. In addition, a silent gene cluster producing annullatins was recently characterized. The main findings are summarized below.

### 4.1. Roquefortine C Biosynthetic Gene Cluster and Pathway

Roquefortine C is produced in *P. roqueforti* by means of a biosynthetic pathway encoded by a gene cluster comprising four genes (*gmt*, *rpt*, *rdh* and *rds*) and one pseudogene [34,46] (Figure 2a, Table 1).

Starting from the *gmt* gene and downstream of it, there is a DNA fragment that shows several in-frame translation stop codons. Upon translation, it would give rise to several protein fragments showing homology to RoqT, an MSF transporter present in the roquefortine C/meleagrin biosynthetic cluster of *P. chrysogenum* that is not essential for the secretion of these metabolites [47]. Due to the organization of this DNA fragment, these peptides seem not to constitute a functional protein. Therefore, this DNA may represent a pseudogene that has not been annotated in the genome of *P. roqueforti* FM164 [48]. Downstream of the pseudogene, the next genes in the cluster are *rpt* (initially dubbed dimethylallyltryptophan synthetase *dmaW*), *rdh,* and *rds.* These last two genes are organized in divergent orientation (head-to-head) and share a divergent promoter (Figure 2a).

The Roquefortine C/meleagrin biosynthetic pathway was first elucidated in *P. chrysogenum* and included a metabolic grid in the early steps and further branching, thus giving rise to different roquefortines, Glandicolines A and B, and meleagrin [47,49,50]. Based on these studies, the Roquefortine C biosynthetic pathway was also characterized in *P. roqueforti* [46]. These two pathways have the initial steps in common and differ in the late steps involved in the formation of Roquefortine L (Glandicoline A) and Glandicoline B (Figure 2b). L-histidine and L-tryptophan are the precursor amino acids of these indole alkaloids [51,52]. They are condensed by the *rds*-encoded dimodular nonribosomal peptide synthetase to form the cyclopeptide histidyltryptophanyldiketopiperazine (HTD), which can be converted either into dehydrohistidyltryptophanyldiketopiperazine (DHTD) by the *rdh*-encoded dehydrogenase, or into Roquefortine D (3,12-dihydro-roquefortine C) by the *rpt*-encoded reverse prenyltransferase, which introduces an isopentenyl group from dimethylallyl diphosphate at the C-3 of the indole moiety. Then, DHTD and Roquefortine D are converted into Roquefortine C by means of the *rpt*-encoded prenyltransferase or by the *rdh*-encoded dehydrogenase, respectively, thus closing the metabolic grid [34,46] (Figure 2b).

*P. roqueforti* cannot produce meleagrin due to the absence of the genes encoding late enzymes for the conversion of Roquefortine C to Roquefortine L (Glandicoline A) (*nox*), and the latter to Glandicoline B (*sro*) that are present in *P. chrysogenum*. However, it contains the *gmt* gene that encodes the methyltransferase catalyzing the methylation of Glandicoline B (yielding meleagrin) and hydroxylated Roquefortine C (yielding roquefortine F) in *P. chrysogenum* [47,49,50]. Therefore, it has been suggested that the *P. roqueforti* methyltransferase encoded by the *gmt* gene could be involved in the conversion of hydroxylated Roquefortine C into minor methylated derivatives (e.g., roquefortine F) [46,53] (Figure 2b).

### 4.2. Isofumigaclavine A Biosynthetic Gene Cluster and Pathway

The identification of a second dimethylallyltryptophan synthase, (*dmaW2*, current annotation: Proq05g069310), different from the roquefortine prenyltransferase (*rpt*, *dmaW* Proq01g022770) involved in the biosynthesis of Roquefortine C, allowed the characterization of the Isofumigaclavine A biosynthetic gene cluster (Figure 3a, Table 2) and elucidation of the biosynthetic pathway in *P. roqueforti* [54] (Figure 3b).

Knock-down mutants in the *dmaW2* gene were able to produce Roquefortine C but were blocked in the biosynthesis of Isofumigaclavine A, thus confirming the role of the protein encoded by *dmaW2* gene in the Isofumigaclavine A biosynthetic pathway of *P. roqueforti.* Consequently, the *dmaW2* gene was renamed *ifgA* (for isofumigaclavine A biosynthesis), and two clusters (A and B) containing the genes putatively involved in the biosynthesis of Isofumigaclavine A were identified in the *P. roqueforti* genome [54,55] (Figure 3a, Table 2). Cluster A includes *ifgE*, *ifgF1*, *ifgD*, *igfB*, *ifgC* and *ifgA* genes, whereas Cluster B contains three genes related to Isofumigaclavine A biosynthesis (*ifgG*, *ifgI* and *ifgF2*) and two extra genes encoding a phytanoyl-CoA dioxygenase (CDM30152) and an unnamed protein product (CDM30154), respectively, of uncharacterized role in this biosynthetic pathway (Figure 3a, Table 2).

The *P. roqueforti* Isofumigaclavine A biosynthetic pathway was proposed [54,55] by comparing the enzymes putatively encoded in Clusters A and B with orthologs of *Neosartorya fumigata*, whose function was characterized by different molecular techniques [56,57]. The formation of dimethylallyltryptophan, specific for clavine biosynthesis, is carried out by the prenyltransferase encoded by *ifgA*. In the next step, the N-methyltransferase encoded by *ifgB* adds a methyl group in the amino group of dimethylallyltryptophan, thereby forming N-methyl-dimethylallyltryptophan. This compound is converted into Chanoclavine I by the coordinated action of the FAD-dependent oxidoreductase (encoded by *ifgC*) and the catalase (encoded by *ifgD*). Then, the short chain dehydrogenase encoded by *ifgE* transforms Chanoclavine I into the aldehyde form, which is converted into festuclavine by the festuclavine synthases (encoded by *ifgF1* and *ifgF2*, the latter located on Cluster B) and the FMN-containing “old yellow enzyme” (encoded by the *ifgG*, located on Cluster B). The next step is catalyzed by the festuclavine hydroxylase, which is a cytochrome P450 that carries out the hydroxylation of C-9, thus forming Isofumigaclavine B. The gene encoding this protein (*ifgH*) is not present in Clusters A or B. Therefore, alternative genes for P450 cytochrome monooxygenases may encode this enzyme. Finally, the acetyltransferase encoded by *ifgI* (located on Cluster B) has been proposed to participate in the conversion of Isofumigaclavine B to Isofumigaclavine A [54,55].

In addition to these enzymes, a second “old yellow enzyme” named FgaOx3Pr3 has been reported in *P. roqueforti,* and the gene encoding this protein is outside Clusters A and B. FgaOx3Pr3 was able to catalyze the formation of festuclavine in the presence of a festuclavine synthase FgaFS [58], thus suggesting that at least two different “old yellow enzymes” may contribute to isofumigaclavine biosynthesis in *P. roqueforti* [55].

### 4.3. PR-Toxin Biosynthetic Gene Cluster and Pathway

The PR-toxin is an aristolochene-derived bicyclic sesquiterpenoid compound of the eremophilane class [27] synthesized by the aristolochene synthase (encoded by *ari1*) from farnesyldiphosphate [59,60]. The *ari1* gene was cloned from *P. roqueforti* [60] and used as reference for the screening of a phage library of *P. roqueforti*, which allowed the identification of four genes (*prx1*-*prx4*) of the PR-toxin biosynthetic pathway, including the *ari1* gene (*prx2*) [61]. This structure was compared to that present in *P. chrysogenum*, thereby revealing that in this microorganism, the *prx* cluster contained 11 genes (Pc12g06260 to Pc12g06370, with Pc12g06290 being a pseudogene) [61]. Later, the availability of the genome sequence of *P. roqueforti* allowed the full characterization of the *prx* gene cluster in this filamentous fungus [34]. Orthologs to *prx1* (Proq02g040180)-*prx11* (Proq02g040260) were identified on scaffolds ProqFM164S02 and ProqFM164S04, since *prx5* (Proq06g077110a, 2233 bp, encoding a putative MFS membrane protein of 567 amino acids), *prx6* (Proq06g077120, 992 bp, encoding a putative NAD-dependent dehydrogenase of 291 amino acids), and *prx7* (Proq06g077130, 1113 bp, encoding a putative NAD-dependent dehydrogenase of 295 amino acids) genes were found on a different genomic region (contig Proq06) away from *prx1*, *prx2* (*ari1*), *prx3*, *prx4*, *prx8*, *prx9*, and *prx11* genes (contig Proq02). In addition, no orthologs were identified for *prx10* in *P. roqueforti*. Instead, two ORFs were found between *prx9* and *prx11* genes and between Proq02g040240 (ORF7) and Proq02g040250 (150 bp, annotated as 49 amino acids unnamed protein product) [34] (Figure 4a). Given this unusual distribution of the *prx* genes in two different genome regions in *P. roqueforti*, a new biosynthetic gene cluster was proposed and also included 11 ORFs, with ORF1 corresponding to the *prx1* gene (Proq02g040180) (Figure 4a, Table 3). This cluster did not include the *prx5*, *prx6,* and *prx7* genes and incorporated three additional genes located downstream of *prx11*: Proq02g040270a (ORF9), Proq02g040280 (ORF10), and Proq02g040290 (ORF 11). This new cluster was similar in terms of gene synteny and protein homologies to that present in *P. chrysogenum* and *Penicillium camemberti*. The main difference was the orientation of ORF7 (Proq02g040240) in *P. roqueforti* regarding the other two species [62].

The silencing of ORF1 to ORF4 (*prx1*-to *prx4* genes) [61] and ORF5, ORF6, and ORF 8 (*prx8*, *prx9* and *prx11* genes) [62], together with the identification of new pathway intermediates (7-epineopetasone) [63], allowed the proposal of hypothetical PR-toxin biosynthetic pathways in *P. roqueforti* (Figure 4b), although the precise function of several enzymes encoded by the *prx* genes and the nature of most pathway intermediates are still unclear.

The PR-toxin is formed by three molecules of isopentenyldiphosphate (converted into farnesyl diphosphate) and from an acetyl group [64]. Farnesyl diphosphate is then cyclized to the 15-carbon molecule aristolochene by the aristolochene synthase, encoded by the *ari1* (*prx2*, ORF2, Proq02g040190) gene [34,59,60,61,62]. Aristolochene is then converted to 7-epineopetasone via an allylic oxidation likely catalyzed by a quinone oxidoreductase encoded by *prx3* (ORF3, Proq02g040200). The formation of Eremofortin B from 7-epineopetasone seems to involve one dehydrogenation (or allylic oxidation), one oxidation, and one epoxidation catalyzed by the monooxygenase encoded by *prx9* (ORF6, Proq02g040230) as confirmed by gene-silencing methods [62]. Initially, it was proposed that Eremofortin B can be acetylated first by the enzyme encoded by *prx11* and further oxidized, with Eremofortin A being a shunt product rather than an intermediate [63]. Later, it was suggested that Eremofortin B undergoes epoxidation by the enzymes encoded by *prx8* (ORF5, Proq02g040220a) or *prx9* (ORF6, Proq02g040230), thereby forming DAC-eremofortin A, the latter being acetylated by the acetyltransferase encoded by *prx11* (ORF8, Proq02g040260) to form Eremofortin A [34]. Finally, it was suggested that Eremofortin A is really an intermediate in the pathway [62], which would be likely oxidized by the dehydrogenase encoded by *prx1* (ORF1, Proq02g040180) and by the monooxygenase encoded by *prx8* (ORF5, Proq02g040220a) to Eremofortin C [34,62] with the formation of Eremofortin D as a byproduct. The last step in the pathway consists of a dehydrogenation at C-12 of Eremofortin C, likely carried out by the protein encoded by *prx4* (ORF4, Proq02g040210) to form the PR-toxin [34].

### 4.4. Mycophenolic Acid Biosynthetic Gene Cluster and Pathway

The information provided in different studies carried out in *Penicillium brevicompactum*, where the mycophenolic acid gene cluster comprising *mpaA*, *mpaB*, *mpaC*, *mpaDE*, *mpaF*, *mpaG,* and *mpaH* genes was identified [65,66,67], laid the foundations for the characterization of the mycophenolic acid biosynthetic gene cluster and pathway in *P. roqueforti*, which showed a similar structure [68,69] (Figure 5a, Table 4).

Starting from the *mpaA* gene, the cluster includes, adjacent to it and in divergent orientation, the *mpaB* gene, which was annotated as a 949-bp gene. However, this gene was predicted and confirmed to be larger (1397 bp) and with an additional exon at the 3′ end [68]. The next gene in the cluster is *mpaC*, and annotated next to it there is an ORF (Proq05g069790) that was not included in the cluster described by Del–Cid et al. [68] and Gillot et al. [69]. The next gene in the cluster is *mpaDE*, which was annotated as a 3710-bp ORF encoding a fusion protein of 932 amino acids made up of cytochrome P450 domain in the N-terminal region and a hydrolase domain in the C-terminal region. However, this gene was predicted and confirmed to be shorter (2929 bp; 788 bp in excess at the 5′ end) [68]. The last three genes in the cluster are *mpaF*, *mpaG,* and *mpaH* (Figure 5a).

Involvement of the *mpa* genes in the biosynthesis of mycophenolic acid in *P. roqueforti* was confirmed by gene-silencing experiments, which led to a reduction in the production of this metabolite after knocking down each of the seven genes [68]. These data, together with early precursor labeling and culture feeding studies [65,66,67,68,70], have led to the proposal of a biosynthetic pathway for mycophenolic acid (Figure 5b). Acetyl-CoA, malonyl-CoA and S-adenosylmethionine are the precursor molecules of mycophenolic acid. These compounds are condensed by the non-reducing polyketide synthase encoded by *mpaC*, thus forming 5-methylorsellinic acid. This molecule is further converted into 4,6-dihydroxy-2-(hydroxymethyl)-3-methylbenzoic acid and 5,7-dihydroxy-4-methylphthalide by sequential reactions carried out by the bifunctional cytochrome P450/hydrolase (lactone synthase) encoded by the *mpaDE* gene. Next, the prenyltransferase encoded by *mpaA* catalyzes the farnesylation of 5,7-dihydroxy-4-methylphthalide, thus forming 6-farnesyl-5,7-dihydroxy-4-methylphthalide, which undergoes oxidative cleavage of the farnesyl chain by the enzyme encoded by *mpaH*, thereby generating demethylmycophenolic acid. The last reaction consists of the methylation of the 5-hydroxy group of the demethylmycophenolic acid to form mycophenolic acid, a reaction that is catalyzed by the O-methyltransferase encoded by *mpaG*. The role of the *mpaB*-encoded protein remains unknown, whereas the probable inosine-5′-monophosphate dehydrogenase encoded by *mpaF* could confer resistance to mycophenolic acid and, in some way, also participate in the production of this metabolite, since knock-down mutants in this gene showed reduced production of mycophenolic acid [68].

### 4.5. Andrastin A Biosynthetic Gene Cluster and Pathway

Following the information published in *P. chrysogenum*, where a ~30-kbp genomic cluster of 11 genes (*adrA*, Pc22g22820 to *adrK*, Pc22g22920) was proposed to encode the enzymes involved in Andrastin A biosynthesis [71], the homologous cluster was characterized in *P. roqueforti* [72]. The main difference with the *P. chrysogenum adr* cluster is the lack of the *adrB* gene in *P. roqueforti*, where only a small fragment of the 5′-end of this gene is present (representing a pseudogene), thereby comprising 10 genes in this filamentous fungus (Figure 6a, Table 5).

Despite the characterization of the *adr* gene cluster, the biosynthetic pathway for Andrastin A biosynthesis could not be elucidated in *P. roqueforti*. Initial studies in *P. chrysogenum* were able to hypothesize the initial steps of this pathway based on similarities of the *adrD*, *adrG*, *adrK*, and *adrH* genes with homologous genes of the fungal austinol and terretonin biosynthetic gene clusters [71]. According to these authors, the iterative Type I polyketide synthase, the prenyltransferase, the methyltransferase, and the FAD-dependent oxidoreductase would consecutively catalyze the four steps leading to the biosynthesis of epoxyfarnesyl-3,5-dimethylorsellinic acid methyl ester from acetyl CoA, malonyl CoA and S-adenosylmethionine. Next, these authors confirmed the function of the protein encoded by *adrI* (terpene cyclase) and were able to complete the biosynthesis of Andrastin A by heterologous co-expression of *adrF*, *adrE*, *adrJ*, and *adrA* genes in a strain of *Aspergillus oryzae* supplemented with Andrastin E. On the other hand, the putative role of the *adrB*- and *adrC*-encoded proteins could not be characterized in *P. chrysogenum* [71].

Given the high percentage of identity (ranging between 83% and 94%) shared by the orthologous *adr*-encoded proteins of *P. chrysogenum* and *P. roqueforti* (with exception of *adrB*, as indicated above), and since the 10 *adr*-encoded proteins are involved in the biosynthesis of andrastin A in *P. roqueforti* (as it was confirmed by RNA-mediated gene silencing of each of the 10 genes of the *adr* gene cluster [72]) a similar pathway to that proposed by *P. chrysogenum* [71] (Figure 6b) is likely to occur in *P. roqueforti*. Interestingly, the ABC transporter encoded by *adrC* (indicated as MFS transporter by [72]), although involved in Andrastin A production in *P. roqueforti*, showed no specific role in the secretion of this metabolite [72].

### 4.6. Annullatins D and F Biosynthetic Gene Cluster and Pathway

Prenylated salicylaldehyde derivatives are composed of a salicylaldehyde scaffold carrying a saturated or an unsaturated C7 side chain [73]. In fungi, prenylated salicylaldehyde derivatives, particularly those belonging to the group of flavoglaucins, have been isolated mainly from the *Aspergillus* (*Eurotium*) species [74,75,76,77,78]. Recently, the nine-gene biosynthetic gene cluster for flavoglaucin (the *fog* cluster) was identified in *Aspergillus ruber* [79]. Taking advantage of this information, Xiang et al. [80] used the *fog* cluster to search for homologues in the genome of *P. roqueforti* FM164. As a result, they identified a ~28-kbp genomic cluster of eleven genes in *P. roqueforti*, the *anu* cluster, which comprised genes *anuA* to *anuK* and included *anuK* and *anuJ* genes at both ends of the cluster (Figure 7a, Table 6).

Based on the putative functions of the *anu* genes, it was speculated that the product of this biosynthetic gene cluster should be a prenylated derivative. To identify the product, the full cluster was cloned in a plasmid and transferred to *A. nidulans* for heterologous expression. The resulting transformant (named *A. nidulans* BK08) was cultivated in a liquid medium. Broths of this transformant were extracted with ethyl acetate and analyzed by LC–MS. As a result, four new products absent in control strain were identified. The structures of these compounds were elucidated by spectroscopic techniques as Annullatin D, Annullatin F, Annullatin G, and Annullatin H [80]. Annullatins are a group of compounds that comprise 2,3-dihydrobenzofurans or aromatic polyketides derivatives and were previously identified in the entomopathogenic fungus *Cordyceps annullata* [81].

Further experiments, including gene deletion of *anu* genes in *A. nidulans* BK08, feeding experiments, and enzymatic characterization allowed for the reconstruction of a putative biosynthetic pathway for Annullatins D and F in *P. roqueforti* [80] (Figure 7b). Acetyl-CoA and malonyl-CoA are the precursor molecules of annullatins. These compounds are condensed to 2-hydroxymethyl-3-pentylphenol by the joint action of a reducing polyketide synthase (encoded by *anuA*), a short-chain dehydrogenases/reductase encoded by *anuB*, and a protein of unknown function encoded by *anuC*, which also seems to be a short-chain dehydrogenases/reductase [80]. Further conversion of 2-hydroxymethyl-3-pentylphenol into Annullatin E is catalyzed by the cytochrome P450 hydroxylase encoded by the *anuE* gene. Next, the prenyltransferase encoded by *anuH* introduces a dimethylallyl group from dimethylallyl diphosphate at the C-6 of phenol moiety of annullatin E thus forming annullatin J. At this point, the dihydrobenzofuran ring formation between the prenyl and the phenolic hydroxyl groups in Annullatin J would give rise to two diastereomers molecules, (*2S*, *9S*)-annullatin H and the hypothetical (*2R*, *9S*)-annullatin H. According to the literature, the formation of this ring could be non-enzymatic or enzymatic [73]. In the case of *anu* cluster, no gene has been involved in this part of the pathway [80]. After the diastereomers formation, the pathway splits into two branches. In the main branch, (*2S*, *9S*)-annullatin H is converted to Annullatin D by two hypothetical sequential reactions carried out by the berberine bridge enzyme-like protein encoded by *anuG*. In the other branch, the hypothetical diastereomer (*2R*, *9S*)-annullatin H is immediately converted to Annullatin F by the short-chain dehydrogenase/reductase encoded by *anuF*.

There are aspects of the *anu* cluster and its biosynthetic pathway that deserve to be highlighted. *P. roqueforti* has never been reported as a producer of annullatins. Indeed, Xiang et al. [80] did not detect annullatins in cultures of *P. roqueforti* FM164 grown under different conditions. These antecedents suggest that the *anu* cluster is silent. However, there is no gene expression data in *P. roqueforti* that could confirm it, but the heterologous expression of the *anu* cluster in *A. nidulans* provides indirect support to this fact (see Section 5.1).

Another interesting aspect is that the precise function of several enzymes encoded by the *anu* cluster and the nature of several pathway intermediates are unclear. Gene deletion experiments of *anu* genes in *A. nidulans* BK08 indicated that *anuD*, *anuI*, and *anuJ* are likely not involved in the annullatins biosynthesis [80]. Therefore, the function (if any) of these genes in *P. roqueforti* is currently unknown. Concerning pathway intermediates, several of them are hypothetical, with isomer (*2R*, *9S*)-annullatin H, from which emerges one of the pathway branches (Figure 7b), being the most important one. It is speculated that this compound is very unstable, and for this reason, it cannot be detected [80].

Finally, there are several compounds that are produced in *A. nidulans* BK08 by shunt pathways, which could not be necessarily produced by natural strains of *P. roqueforti*. The main shunt pathway is observed after the deletion of *anuE* gene in *A. nidulans* BK08. Under these conditions, 2-hydroxymethyl-3-pentylphenol would be transformed to a hypothetical prenylated derivative by the prenyltransferase encoded by *anuH*. This prenylated precursor would be subsequently metabolized to Annullatin I, Annullatin B, and Annullatin A by unknown mechanisms (Figure 7b). Another shunt product, Annullatin G, would be produced from Annullatin F through an enzymatic reaction catalyzed by an unidentified endogenous enzyme from *A. nidulans* [80] (Figure 7b).

## 5. Control and Regulation of the Biosynthesis of Secondary Metabolites in *P. roqueforti*

The fungal secondary metabolism is controlled by a wide array of regulators, which lead directly or indirectly to the activation or repression of a given biosynthetic gene cluster. Depending on the level and extension of the effect, regulators of fungal secondary metabolism can be classified in two main groups: cluster-specific regulators and global regulators [82]. A cluster-specific regulator (also named specific transcription factor or narrow domain regulator) refers to a regulatory protein with a high similarity to a transcription factor that is encoded by a gene that is part of a specific biosynthetic gene cluster. More importantly, this transcription factor specifically regulates the expression of the other genes of the biosynthetic gene cluster [82]. On the contrary, global regulators (also named wide domain regulators) are encoded by genes located outside of a biosynthetic gene cluster and are pleiotropic regulators in fungi. Although most of these regulators are transcription factors, this category also includes signal transducer proteins and epigenetic regulators [82]. In the following sections, we will summarize the current knowledge about the regulation of biosynthetic gene clusters in *P. roqueforti* by cluster-specific and global regulators.

### 5.1. Cluster-Specific Regulators in Biosynthetic Gene Clusters Functionally Characterized in P. roqueforti

As mentioned before, the biosynthetic gene clusters for the biosynthesis of PR-toxin, Andrastin A, Roquefortine C, mycophenolic acid, annullatins, and Isofumigaclavine A have been functionally characterized in *P. roqueforti* [46,54,61,62,68,69,72,80]. Among them, only the gene clusters for the biosynthesis of annullatins and PR-toxin contain candidate genes for a specific transcription factor [62,80].

In the case of the biosynthetic gene cluster of Annullatin D, the functionality of the gene encoding the specific transcription factor (named *anuK*) was partially demonstrated [80]. As mentioned in Section 4.6, the Annullatin D biosynthetic gene cluster seems to be silent. Therefore, it was heterologously expressed in *A. nidulans* for its functional characterization. For this purpose, the biosynthetic gene cluster was previously cloned into plasmid pJN017, which contains the constitutive promoter of the *gpdA* gene. Therefore, the authors took advantage of the fact that the first gene of the biosynthetic gene cluster corresponds to *anuK* (Figure 7) and cloned it without the natural promoter of *anuK* in such a way that this gene was under the control of the *gpdA* promoter [80]. As a result, the production of Annullatin D, Annullatin F, and other related metabolites was achieved, as described in Section 4.6. At this point, it should be noted that beyond this result, there is no further experimental evidence confirming the role of AnuK as the transcriptional activator of the annullatin biosynthetic gene cluster in *P. roqueforti*. However, bioinformatic analysis supports such a role. AnuK has high similarity to FogI, a putative transcription factor found in the biosynthetic gene cluster for the biosynthesis of flavoglaucin in *A. ruber* [79]. In addition, AnuK contains a Zn(II)_2_Cys_6_ domain. This kind of domain is found almost exclusively in fungi and has been linked to several functions, including the regulation of the secondary metabolism [83,84].

Concerning the biosynthetic gene cluster of PR-toxin, its functional characterization in *P. roqueforti* was reported in two consecutive papers [61,62], where seven genes were silenced by RNAi-mediated gene-silencing technology. Unfortunately, the gene encoding the putative transcription factor of the biosynthetic gene cluster, named ORF10 (Figure 4) [62], was not analyzed. Similar to AnuK described before, ORF10 encodes a protein that contains a Zn(II)_2_Cys_6_ domain. ORF10 is highly similar to orthologous genes found in the biosynthetic gene clusters for PR-toxin in *P. camemberti* and *P. chrysogenum* [62]. Thus, the high degree of conservation of ORF10 in biosynthetic gene clusters of different fungi suggests that this gene could be functional. In the future it will be interesting to determine the role of ORF10 in the regulation of the biosynthetic gene cluster for the biosynthesis of PR-toxin in *P. roqueforti*.

### 5.2. Global Regulators of Biosynthetic Gene Clusters in P. roqueforti

Almost all the current knowledge about the regulation of biosynthetic gene clusters in *P. roqueforti* corresponds to global regulators. However, it should be kept in mind that in comparison with model fungi (*Aspergillus* spp., *P. chrysogenum*, etc.), this knowledge is still scarce. Indeed, as far as we know, the effect of only three global regulators on the secondary metabolism have been studied in *P. roqueforti*. They are detailed below.

#### 5.2.1. The *pga1* Gene Encoding for an α-Subunit of a Heterotrimeric G Protein

The first global regulator studied in *P. roqueforti* was a heterologous gene named *pga1*, which encodes an α-subunit of a heterotrimeric G protein. Heterotrimeric G proteins, particularly α-subunits, are important in fungi and have been involved mainly in the regulation of growth, asexual reproduction, and secondary metabolism [85]. García–Rico et al. [86] isolated and characterized the *pga1* gene from *P. chrysogenum*, and they decided to perform the heterologous expression of a mutant version of this gene in *P. roqueforti* [87]. For this purpose, they used a plasmid containing a mutant version of *pga1* (named *pga1^G42R^*), which encodes a protein where a glycine is replaced by arginine at Position 42, producing a “constitutively active” Pga1 protein that is always signaling [88]. The introduction of this “constitutive” allele in *P. roqueforti* produced several phenotypic effects. In the case of secondary metabolism, García–Rico et al. [87] measured the production of Roquefortine C during 30 days in a strain of *P. roqueforti* containing the *pga1^G42R^* allele and observed drastic changes in the production of this secondary metabolite as compared with the wild-type strain. Namely, the strain containing *pga1^G42R^* increased production of the mycotoxin, reaching 0.7 μg/mg of dry mycelium at day 18, whereas the wild-type strain produced 0.4 μg/mg of dry mycelium at the same day. Interestingly, the levels of Roquefortine C were higher in the *pga1^G42R^* strain throughout a culture period ranging between 10–30 days [87]. These results suggested that *pga1* has a positive effect on the production of Roquefortine C in *P. roqueforti* [87]. As mentioned above, these experiments were performed using a heterologous gene from *P. chrysogenum*. Therefore, the role of the native *pga1* gene on the secondary metabolism of *P. roqueforti* remains to be tested.

#### 5.2.2. The *sfk1* Gene Encoding for Suppressor of Four-Kinase 1 Protein

Another global regulator influencing the biosynthesis of secondary metabolites in *P. roqueforti* is Sfk1 (Suppressor of four-kinase 1), a transmembrane protein located on the plasma membrane that was originally described in *Saccharomyces cerevisiae* [89]. In this yeast, Sfk1 physically interacts with Stt4, an important phosphatidylinositol 4-kinase involved in the phosphoinositide second messenger’s pathway [90]. Stt4 must be localized to the plasma membrane to fulfill its role. Hence, it is thought that its interaction with Sfk1 would allow its proper subcellular localization [91]. In addition, Sfk1 is essential for the retention of ergosterol in the plasma membrane of the yeast [92]. The mammalian homologue of Sfk1 was also studied [93,94]. The role of this homologue is essentially similar to that described in yeast, although it does not have a direct role in mammals in the localization of the phosphatidylinositol 4-kinase to the plasma membrane [93]. In addition, it has been suggested that in mammals, Sfk1 would also be a negative regulator of transbilayer movement of phospholipids in plasma membrane [94].

The interest in studying Sfk1 in *P. roqueforti* emerged from a previous work by Gil-Durán and co-workers [95]. These authors performed suppression subtractive hybridization (SSH) experiments, looking for downstream genes regulated by Pga1. Among the sequences obtained from these experiments, the full cDNA of *sfk1* gene was found. Considering this antecedent, Torrent and co-workers [96] decided to analyze the role of this gene in *P. roqueforti*. For this purpose, they performed the RNA-mediated gene-silencing of *sfk1* and measured several biological properties in the transformants. In the case of secondary metabolites, they observed that the knock-down of *sfk1* produced a drastic decrease in the production of Roquefortine C, Andrastin A, and mycophenolic acid in *P. roqueforti* [96]. These results suggest that *sfk1* is a positive regulator of the production of these three secondary metabolites in *P. roqueforti*.

#### 5.2.3. The *pcz1* Gene Encoding a Protein with a Zn(II)_2_Cys_6_ Domain

The last global regulator of secondary metabolism studied in *P. roqueforti* is that encoded by *pcz1* (Penicillium C6 zinc-finger protein 1), a gene whose role in fungi was unknown until recent time. This gene encodes a protein containing a Zn(II)_2_Cys_6_ domain [95]. As in the case of *sfk1*, *pcz1* was also obtained from SSH experiments performed on *P. roqueforti* [95]. Since there are orthologues of this gene in all ascomycetes analyzed so far [95], it seemed interesting to determine its role on the secondary metabolism of *P. roqueforti*. For this purpose, two kinds of *P. roqueforti* strains were obtained: strains overexpressing *pcz1* [97] and strains where *pcz1* was knocked-down by RNA-silencing technology [95]. Using both types of strains, the production of Roquefortine C, Andrastin A, and mycophenolic acid was measured in comparison to the wild-type strain [97]. In the case of mycophenolic acid, a clear effect was found. Strains overexpressing *pcz1* showed higher titers of mycophenolic acid than the wild-type strain, which correlated with higher levels of the expression of key genes from the mycophenolic acid biosynthetic gene cluster [97]. On the contrary, strains where *pcz1* was knocked down produced lower levels of the mycophenolic acid as compared to wild-type fungus in concomitance with lower levels of the expression of genes from its biosynthetic gene cluster [97]. These results indicate that *pcz1* exerts a positive control of the production of mycophenolic acid and suggest that this effect is mediated by modifying the transcriptional status of the biosynthetic gene cluster of mycophenolic acid.

Unlike mycophenolic acid, important reductions in the production of Roquefortine C and Andrastin A were observed in both types of strains [97]. These results were confirmed by measuring the level of transcription of key genes from Roquefortine C and Andrastin A biosynthetic gene clusters [97]. Thus, the effect of *pcz1* on these metabolites was unexpected and difficult to interpret. Considering these results, an indirect effect of *pcz1* on the production of Roquefortine C and Andrastin A was suggested, and some hypotheses were proposed. The first one was related to competition for the use of sharing substrates. These three compounds require acetyl CoA directly or indirectly for their biosynthesis. Therefore, the overexpression of the mycophenolic acid pathway (produced in turn by the overexpression of *pcz1*) would use more efficiently acetyl-CoA, in detriment of the production of Andrastin A and Roquefortine C. This would produce a “negative loop” in the expression of Andrastin A and Roquefortine C biosynthetic gene clusters, thereby resulting in a low production of these compounds. The second hypothesis was the alteration of the normal balance of unknown regulators of Roquefortine C and Andrastin A because of the overexpression of *pcz1* [97]. A similar phenomenon was described before in *Aspergillus*. In this case, it was observed that the elimination of the regulator *mtfA* decreased the production of two related mycotoxins: aflatoxin and sterigmatocystin [98,99]. However, the overexpression of *mtfA* also decreased the production of these mycotoxins because *mtfA* downregulates the normal expression of the transcriptional factor *aflR*, which is a positive activator of the biosynthesis of aflatoxin and sterigmatocystin in Aspergillus [98,99]. Finally, the last hypothesis suggested the existence of a regulatory mechanism not fully understood, known as regulatory cross-talk [97]. Regulatory cross-talk refers to an inter-regulation amongst different and non-related biosynthetic gene clusters in fungi [100]. Regulatory cross-talk was originally described in *A. nidulans*, where the overexpression of a gene encoding a NRPS of the so-called *inp* biosynthetic gene cluster resulted in the unexpected activation of the non-related biosynthetic gene cluster for the biosynthesis of asperfuranone [101].

In the literature, there are several other examples of regulatory cross-talk in different fungi [100]. In *P. roqueforti*, regulatory cross-talk has been described in the case of biosynthetic gene clusters encoding PR-toxin and mycophenolic acid [61]. More precisely, during the experiments of the functional characterization of the PR-toxin biosynthetic gene cluster [61], it was noticed that the down regulation of several genes of this biosynthetic gene cluster by RNAi-mediated silencing technology largely increased the production of mycophenolic acid. It is important to highlight that PR-toxin and mycophenolic acid are unrelated secondary metabolites, so this result can hardly be due to an imbalance of shared substrates. PR-toxin is a bicyclic sesquiterpene that belongs to eremophilane. Its core structures is aristolochene, a sesquiterpene produced from farnesyl diphosphate [28], whereas mycophenolic acid is a meroterpenoid whose core structure, 5-methylorsellinic acid (5-MOA), is formed by a polyketide synthase. 5-MOA is subjected to several further modifications (including prenylation) to yield mycophenolic acid [102]. Thus, both compounds are synthesized by entirely different biosynthetic pathways in *P. roqueforti* [62,68].

Considering the previous observation of regulatory cross-talk in *P. roqueforti*, Rojas–Aedo and co-workers [97] suggested that the unexpected result in the regulation of mycophenolic acid, Andrastin A and Roquefortine C by *pcz1* in this fungus, could be due to the existence of regulatory cross-talk between the biosynthetic gene clusters of these metabolites. Hence, when *pcz1* is subjected to genetic manipulation, a complex network of regulatory cross-talk between these biosynthetic gene clusters is triggered. Depending on the case, the overproduction or reduction in the levels of some compounds is observed. However, this remains as a hypothesis that requires further experimental support.

### 5.3. Concluding Remarks on Regulation of Secondary Metabolism in P. roqueforti

In comparison with other fungi, our current knowledge about the regulation of the biosynthesis of secondary metabolites in *P. roqueforti* is poor, which should encourage fungal biologists to pay more attention to this interesting topic. As mentioned above, to date, barely one specific regulator of a biosynthetic gene cluster has been indirectly characterized in *P. roqueforti* [80], and some important global regulators, such as CreA, LaeA, PacC, or AreA have not yet been analyzed in this fungus. This is unlike in other fungi, where their role in the control of secondary metabolism has been largely demonstrated. For example, CreA exerts carbon catabolic repression on penicillin biosynthesis and the expression of the *pcbAB* (the gene encoding the first enzyme of the penicillin pathway) in *P. chrysogenum* [103], while PacC exerts pH-dependent control on the production of patulin and the expression of its BGC in *P. expansum* [104]. In the case of LaeA, its influence on the secondary metabolism of several Penicillium species has been documented [82]. Concerning the regulator of the nitrogen metabolism AreA, it has been involved in the control of the production of secondary metabolites in *P. chrysogenum* and *P. griseofulvum* [82]. Interestingly, the *areA* gene homologue has been studied in *P. roqueforti* [105], but its role on secondary metabolism has not been addressed in this fungus.

## 6. Future Challenges and Perspectives in the Study of Secondary Metabolism in *P. roqueforti*

*P. roqueforti* provides cheese with characteristic organoleptic properties. These properties have been associated to the potent proteolytic and lipolytic activities of this fungus [20,21,106]. In the case of the secondary metabolites produced by *P. roqueforti*, it remains to be determined if they have any role in the organoleptic properties of the cheeses. It is known that *P. roqueforti* inoculates secondary metabolites to cheese in which it grows [42,107]. Therefore, one of the challenges to be addressed in this fungus is whether these secondary metabolites confer organoleptic properties.

It has been estimated that the number of fungal species in nature is between 2- and 11-million species, of which around 150,000 are formally described taxa [108]. Therefore, most of the biosynthetic potential of fungi is yet to be discovered. On the other hand, genomic analyses performed on known fungal species indicate that 80% of their metabolic potential remains unknown [2]. In *Penicillium* species, a large proportion of its biosynthetic gene clusters, in some cases up to an astonishing 90%, have not been connected yet to any molecule [8]. These data indicate that in fungi, including *Penicillium* species, secondary metabolism and biosynthetic gene clusters have been hitherto barely investigated. This is also true for *P. roqueforti*. Recent analyses of several genomes from *P. roqueforti* revealed that depending on the strain, they contain between 34–37 biosynthetic gene clusters [10,109]. As it has been detailed in this review, to date, only six of these biosynthetic gene clusters have been experimentally linked to secondary metabolites produced by *P. roqueforti*. Therefore, the rest of the biosynthetic gene clusters (between 28–31, depending on the strain, representing around 80% of the total biosynthetic gene clusters of *P. roqueforti*), have not been hitherto associated to any known compound. It would be also interesting in the future to study the expression of those biosynthetic gene clusters when *P. roqueforti* or related species are grown on cheese as a substrate.

One of the most interesting research questions to be addressed in *P. roqueforti* is the assignment of a biosynthetic gene cluster of unknown function to a specific secondary metabolite. To answer this question, the coupling of microbiological and molecular biology techniques with chemical methodologies is necessary [110,111,112,113]. This strategy requires tools for efficient genetic manipulation of the fungal biosynthetic gene clusters of interest. Gene deletion has been the gold standard for this purpose. Gene deletion can be performed by gene targeting; that is, the replacement of a gene of a biosynthetic gene cluster with a non-functional version of the same gene through random events of homologous recombination [110,114,115]. Unfortunately, in filamentous fungi, gene targeting is a cumbersome process. In most filamentous fungi, DNA integration is mainly driven by non-homologous recombination, whereas homologous recombination events occur at frequencies between 1–5%, or even less [116,117]. As a result, successful gene-targeted deletion events occur at such a low frequency that it makes this technique impractical for many fungi [118]. In the case of *P. roqueforti*, there is only one report of successful gene deletion, [119] which, to the best of our knowledge, has not been replicated by other researchers.

The lack of efficient tools for gene knock-out in *P. roqueforti* has led to most of the studies of its biosynthetic gene clusters having been done by using RNA-mediated gene-silencing (or gene knock-down). Gene silencing affects gene functionality by reducing its expression but does not affect gene structure (the gene remains intact within the fungal genome) [120,121]. Gene silencing has high efficiency in fungi [121,122]. In some cases, up to 95% of reduced gene expression has been observed, thus yielding phenotypes close to knock-out strains [68,123,124]. Due to its simplicity (this technique only requires the expression of a very small double-stranded RNA molecule [121,122]), gene silencing allows the quick analysis of several genes of a biosynthetic gene cluster at the same time. Considering all these advantages, RNA-mediated gene-silencing has become the most useful technique for the study of biosynthetic gene clusters in *P. roqueforti* [46,54,61,62,68,69,72].

The scenario described above is now changing thanks to the development of CRISPR–Cas9 systems. In past years, several CRISPR–Cas9 systems dedicated to filamentous fungi have been developed and successfully applied to inactivate biosynthetic gene clusters, thus opening new horizons in the research of fungal secondary metabolism [114,125]. Recently, Seekles and co-workers [126] developed the first CRISPR–Cas9 system dedicated to *P. roqueforti*, which has already been successfully applied to the inactivation of genes in this fungus [126,127]. We believe that the availability of this CRISPR/Cas9 system will be a decisive boost for the study of biosynthetic gene clusters, as well as regulatory genes in *P. roqueforti*. This technique must be the starting point to the “synthetic biology era” in this fungus, thereby allowing the re-programming of *P. roqueforti* to produce novel secondary metabolites in ways that have not yet been explored.

## Figures and Tables

**Figure 1 jof-09-00459-f001:**
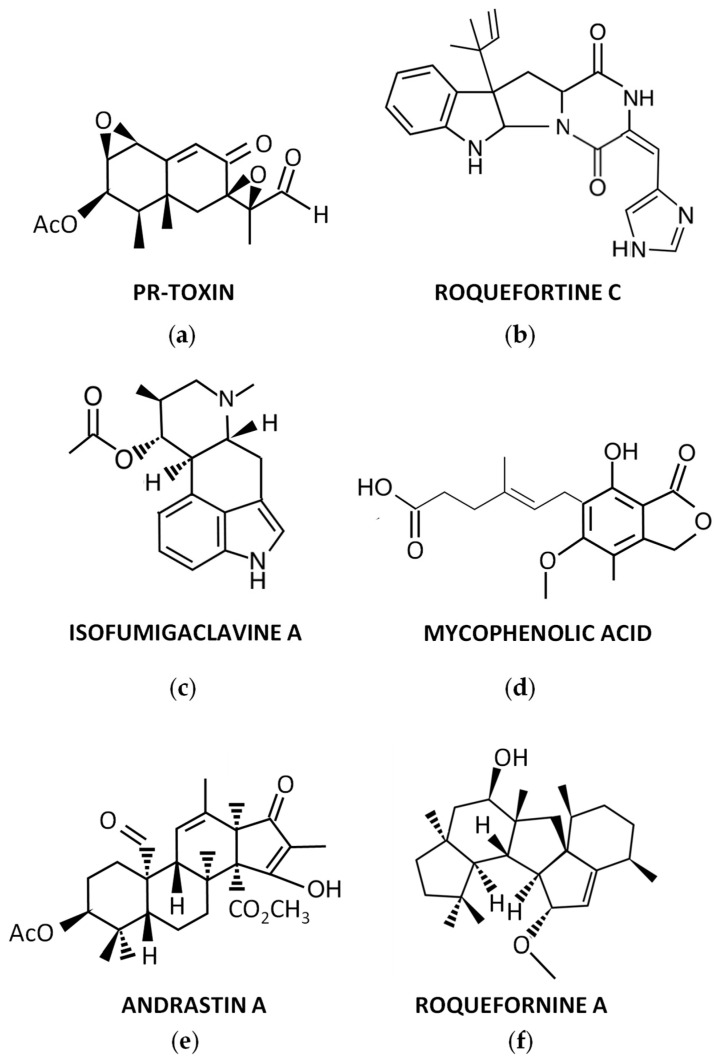
Representative secondary metabolites produced by *P. roqueforti*: (**a**) PR-toxin; (**b**) Roquefortine C; (**c**) Isofumigaclavine A; (**d**) Mycophenolic acid; (**e**) Andrastin A; (**f**) Roquefornine A.

**Figure 2 jof-09-00459-f002:**
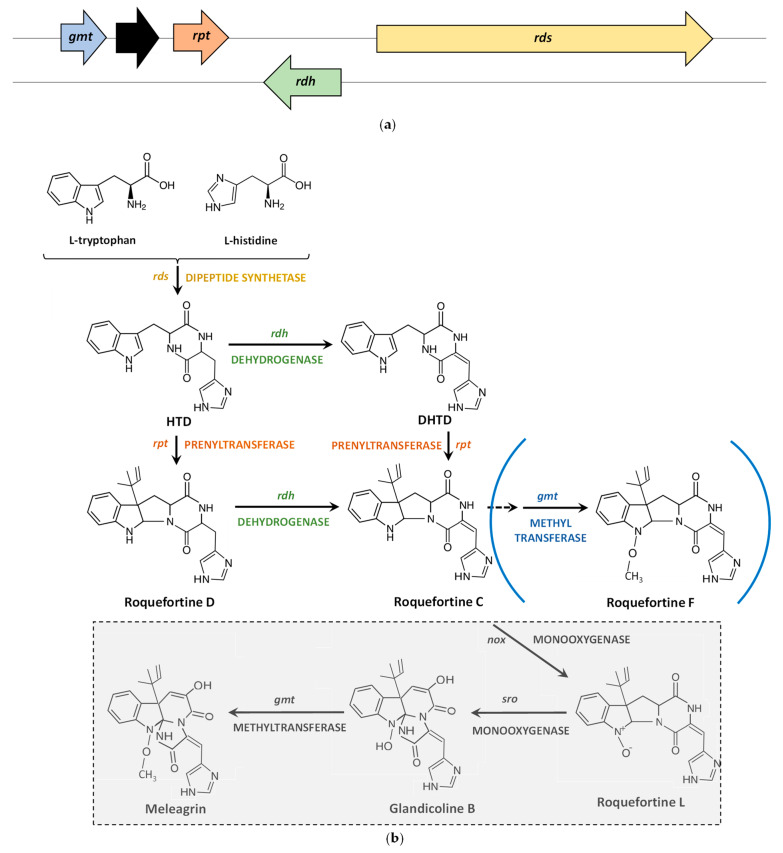
Biosynthesis of Roquefortine C/meleagrin: (**a**) Roquefortine C biosynthetic gene cluster in *P. roqueforti*. The pseudogene is indicated as a black box; (**b**) Roquefortine C/meleagrin biosynthetic pathway in *P. roqueforti* and *P. chrysogenum*. The late reactions only present in *P. chrysogenum* are included within a shaded box, whereas side reactions suggested to involve the *gmt*-encoded methyltransferase and leading to the formation of minor methylated derivatives (e.g., roquefortine F) in *P. roqueforti*, are indicated between blue parentheses. Adapted from [46].

**Figure 3 jof-09-00459-f003:**
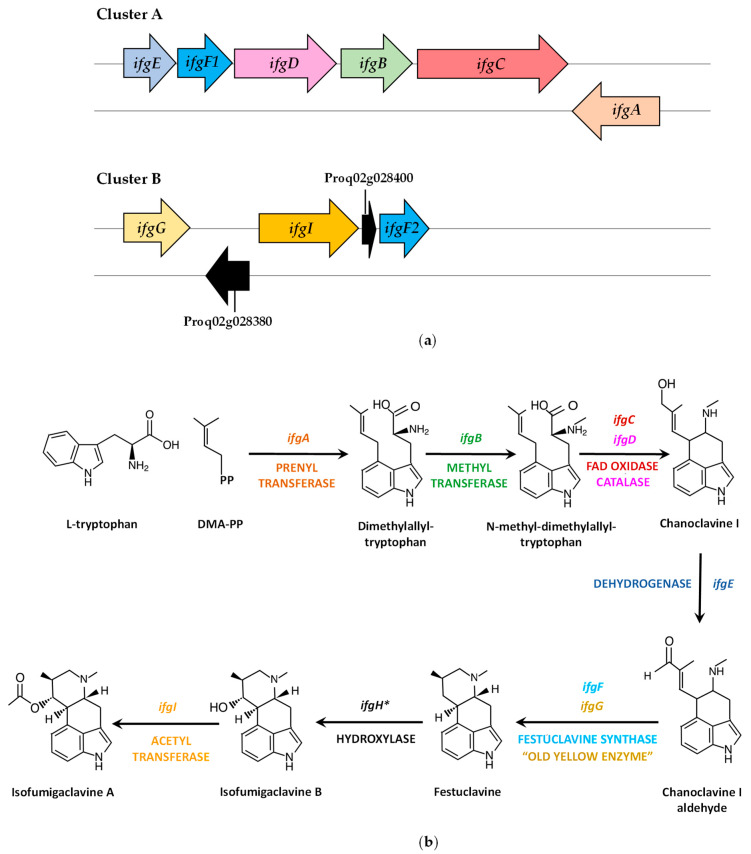
Biosynthesis of isofumigaclavine A in *P. roqueforti*: (**a**) Isofumigaclavine A biosynthetic gene cluster A and B in *P. roqueforti*. Those genes encoding proteins with uncharacterized role in this biosynthetic pathway (Proq02g028380 and Proq02g028400) are indicated with black arrows; (**b**) Isofumigalavine A biosynthetic pathway in *P. roqueforti*. The festuclavine synthase activity corresponds to the activity of two duplicated proteins encoded by *ifgF1* and *igF2* (represented as *ifgF*). The *ifgH* gene encoding the festuclavine hydroxylase is not located in Cluster A or B, and it is denoted with an asterisk. DMA-PP (dimethylallyl pyrophosphate); PP (pyrophosphate). Adapted from [54,55].

**Figure 4 jof-09-00459-f004:**
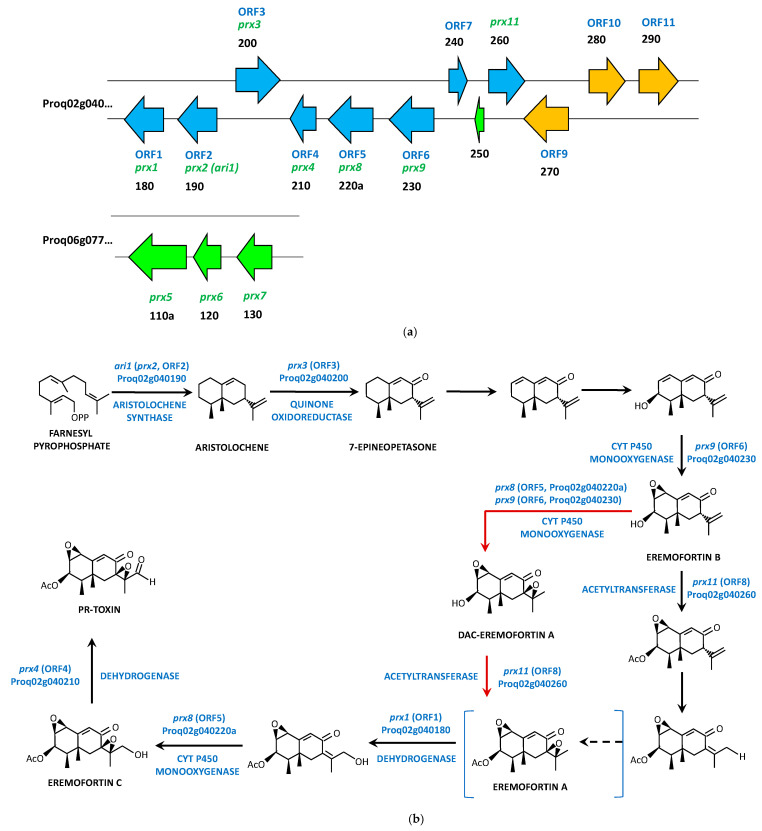
Biosynthesis of PR-toxin: (**a**) PR-toxin biosynthetic gene cluster in *P. roqueforti*. The cluster initially described by García–Estrada and Martín [34] includes genes represented as green (contig Proq06) and blue arrows contig (Proq02), whereas the cluster later described by Hidalgo et al., 2017 includes genes represented as blue and orange arrows (ORF 1 to ORF 11). The nomenclature of each gene, according to its position in each contig, is also indicated in black color; (**b**) Proposed biosynthetic pathway for PR-toxin in *P. roqueforti*. The alternative steps for the transformation of Eremofortin B to Eremofortin A are indicated with red arrows. Eremofortin A has been suggested to be either a shunt product or an intermediate (in blue brackets) adapted from [34,61,62].

**Figure 5 jof-09-00459-f005:**
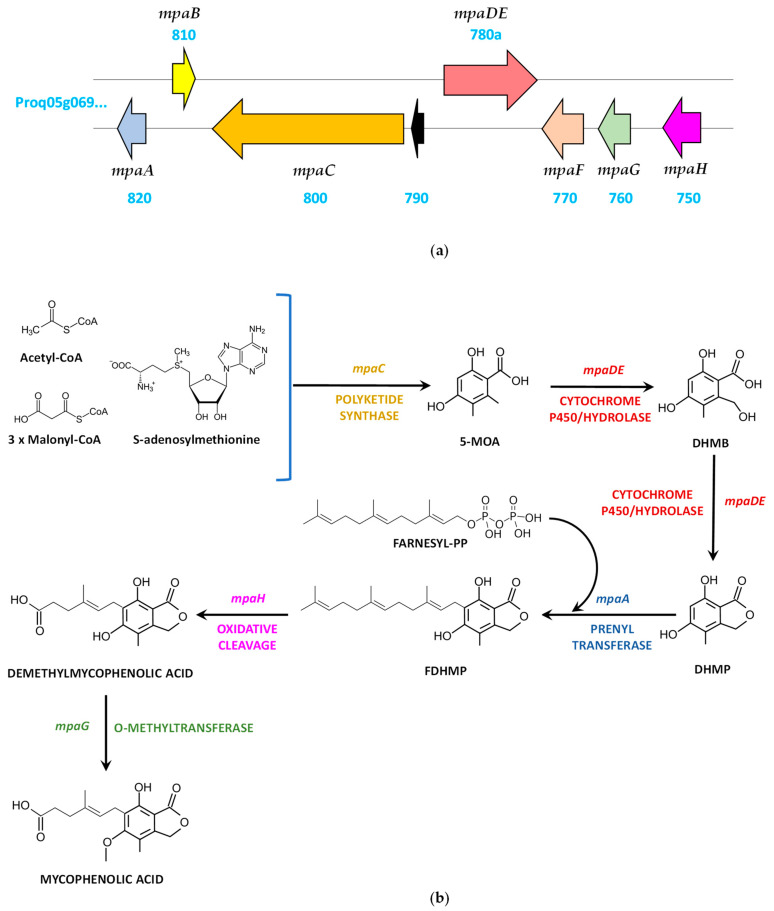
Biosynthesis of mycophenolic acid: (**a**) Mycophenolic acid biosynthetic gene cluster in *P. roqueforti*.; (**b**) Proposed biosynthetic pathway for mycophenolic acid. 5-MOA (5-methylorsellinic acid); DHMB (4,6-dihydroxy-2-(hydroxymethyl)-3-methylbenzoic acid); DHMP (5,7-dihydroxy-4-methylphthalide); FDHMP (6-farnesyl-5,7-dihydroxy-4-methylphthalide); PP (pyrophosphate). Adapted from [34,67].

**Figure 6 jof-09-00459-f006:**
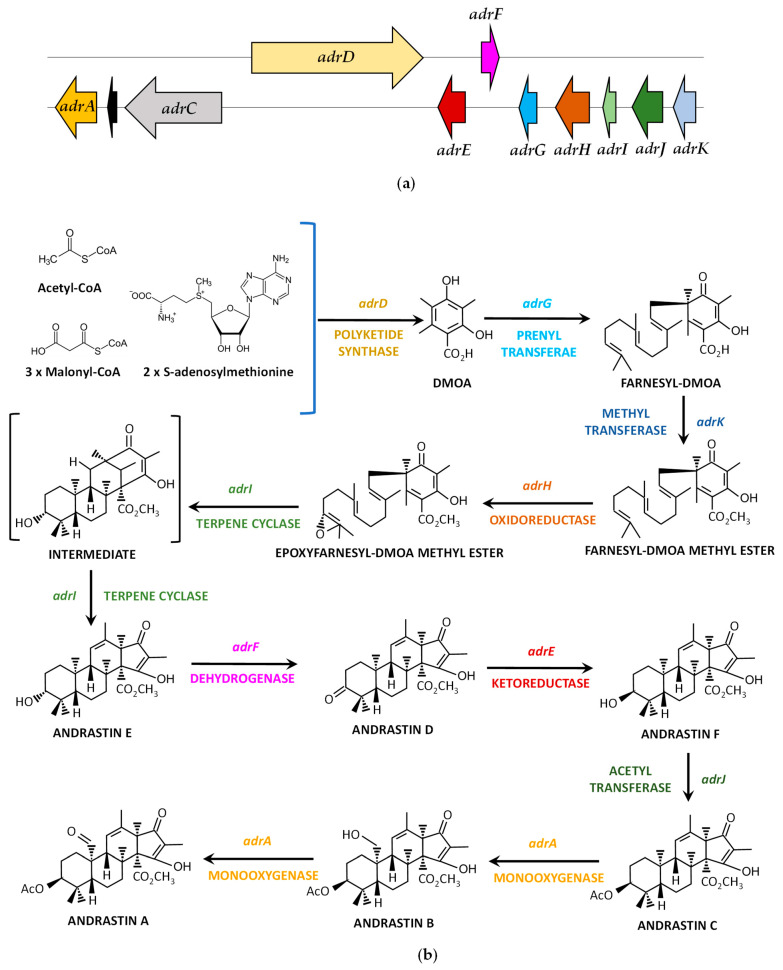
Biosynthesis of Andrastin A: (**a**) Andrastin A biosynthetic gene cluster in *P. roqueforti*. The pseudogene is indicated as a black box; (**b**) Proposed biosynthetic pathway for Andrastin A in *P. chrysogenum*. Likely, this pathway is similar in *P. roqueforti*. DMOA (3,5-dimethylorsellinic acid). Adapted from [71,72].

**Figure 7 jof-09-00459-f007:**
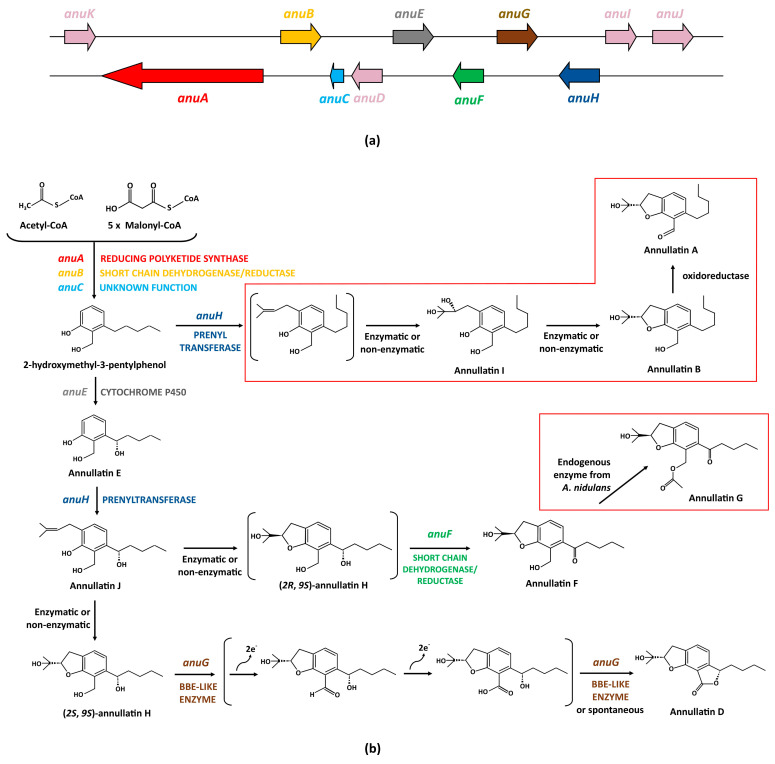
Biosynthesis of Annullatins D and F: (**a**) Annullatin biosynthetic gene cluster in *P. roqueforti*. The functions of *anuD*, *anuI,* and *anuG* remain unknown. The *anuK* gene, encoding a hypothetical transcription factor, is discussed below; (**b**) Proposed biosynthetic pathway for Annullatins D and F in *P. roqueforti*. Hypothetical intermediates are indicated between parentheses. Shunt pathways observed in *Aspergillus nidulans* are included within red boxes. Adapted from [80].

**Table 1 jof-09-00459-t001:** Roquefortine C biosynthetic gene cluster in *P. roqueforti*.

Gene	ORF Name	Size (bp)	Protein	Size (aa)	Proposed Function
*gmt*	Proq01g022760	1058	Methyltransferase	332	Formation of methylated derivatives
Pseudogene	_	_	_	_	_
*rpt* (*dmaW*)	Proq01g022770	1337	Reverse prenyltransferase	425	Addition of an isopentenyl group to the cyclopeptide
*rdh*	Proq01g022780	1638	Dehydrogenase	517	Dehydrogenation of the cyclopeptide
*rds*	Proq01g022790	7200	Nonribosomal peptide synthetase	2363	Formation of the cyclopeptide from L-histidine and L-tryptophan

**Table 2 jof-09-00459-t002:** Isofumigaclavine A biosynthetic gene clusters (A) and (B) in *P. roqueforti*.

**Gene (A)**	**ORF Name**	**Size (bp)**	**Protein**	**Size (aa)**	**Proposed Function**
*ifgE*	Proq05g069260	848	Short-chain dehydrogenase/reductase (CDM36673)	261	Formation of chanoclavine I aldehyde
*ifgF1*	Proq05g069270	992	Festuclavine synthase (CDM36674)	287	Contribution to the formation of festuclavine
*ifgD*	Proq05g069280	1464	Catalase (CDM36675)	466	Contribution to the formation of chanoclavine I
*ifgB*	Proq05g069290	1087	SAM-dependent methyltransferase (CDM36676)	340	Formation of N-methyl-dimethylallyltryptophan
*ifgC*	Proq05g069300	1953	FAD oxidase (CDM36677)	629	Contribution to the formation of chanoclavine I
*ifgA*	Proq05g069310	1507	Dimethylallyltryptophan synthase (CDM36678)	462	Formation of dimethylallyltryptophan
**Gene (B)**	**ORF Name**	**Size (bp)**	**Protein**	**Size (aa)**	**Proposed Function**
*ifgG*	Proq02g028370	1128	FMN-containing “old yellow enzyme” aldolase-type protein (CDM30151)	375	Contribution to the formation of festuclavine
*-*	Proq02g028380	861	Phytanoyl-CoA dioxygenase (CDM30152)	286	Unknown
*ifgI*	Proq02g028390	1455	acetyltransferase (CDM30153)	484	Conversion of isofumigaclavine B to isofumigaclavine A
*-*	Proq02g028400	511	Unnamed protein product (CDM30154)	121	Unknown
*ifgF2*	Proq02g028410	996	Festuclavine synthase (CDM30155)	287	Contribution to the formation of festuclavine

**Table 3 jof-09-00459-t003:** PR-toxin biosynthetic gene cluster in *P. roqueforti* as proposed by [62].

Gene	ORF Name	Size (bp)	Protein	Size (aa)	Proposed Function
*prx1* (ORF1)	Proq02g040180	1072	Short-chain dehydrogenase	340	Oxidation of eremofortin A
*prx2* (*ari1*) (ORF2)	Proq02g040190	1129	Aristolochene synthase	342	Biosynthesis of aristolochene
*prx3* (ORF3)	Proq02g040200	1716	Quinone oxidase	521	Formation of 7-epineopetasone
*prx4* (ORF4)	Proq02g040210	990	Short chain alcohol dehydrogenase	329	Dehydrogenation of eremofortin C
*prx8* (ORF5)	Proq02g040220a	1827	Cytochrome P450 monooxygenase	540	Oxidation of intermediates
*prx9* (ORF6)	Proq02g040230	1792	Cytochrome P450 monooxygenase	579	Oxidation of intermediates
ORF7	Proq02g040240	613	Outer membrane protein, beta-barrel	186	Unknown
*prx11* (ORF8)	Proq02g040260	1416	Acetyltransferase	471	Acetylation of intermediates
ORF9	Proq02g040270a	1823	Cytochrome P450 monooxygenase	509	Oxidation of non-activated hydrocarbons
ORF10	Proq02g040280	1437	Transcriptional regulator	458	Regulation of the gene cluster transcription
ORF11	Proq02g040290	1650	Cytochrome P450 monooxygenase	480	Oxidation of non-activated hydrocarbons

**Table 4 jof-09-00459-t004:** Mycophenolic acid biosynthetic gene cluster in *P. roqueforti*.

Gene	ORF Name	Size (bp)	Protein	Size (aa)	Proposed Function
*mpaA*	Proq05g069820	1126	Prenyltransferase	325	Farnesylation of 5,7-dihydroxy-4-methylphthalide
*mpaB*	Proq05g069810	949 (1397)	Protein of unknown function	298 (427)	Unknown
*mpaC*	Proq05g069800	7715	Non-reducing polyketide synthase	2477	Formation of 5-methylorsellinic acid from acetyl-CoA, malonyl-CoA and S-adenosylmethionine
-	Proq05g069790	523	Protein of unknown function	116	Unknown
*mpaDE*	Proq05g069780a	3710 (2929)	Fusion protein (Cytochrome P450/hydrolase)	932 (852)	Sequential hydroxylation/lactonization of 5-methylorsellinic acid
*mpaF*	Proq05g069770	1708	Inosine-5′-monophosphate dehydrogenase like	526	Role in self-resistance to mycophenolic acid and in the production of this compound
*mpaG*	Proq05g069760	1250	O-methyltransferase	398	Methylation of demethylmycophenolic acid to form mycophenolic acid
*mpaH*	Proq05g069750	1479	Unnamed protein product	433	Oxidative cleavage of the farnesyl chain to form demethylmycophenolic acid

**Table 5 jof-09-00459-t005:** Andrastin A biosynthetic gene cluster in *P. roqueforti*.

Gene	ORF Name	Size (bp)	Protein	Size (aa)	Proposed Function
*adrA*	Proq04g062820	1746	Cytochrome P450 monooxygenase	508	Consecutive oxidations for the formation of Andrastin A from Andrastin C
*adrC*	Proq04g062830	4585	ABC transporter	1452	Unknown (somehow involved in the production of Andrastin A)
*adrD*	Proq04g062840	7973	Polyketide synthase	2495	Formation of 3,5-dimethylorsellinic acid from acetyl-CoA, malonyl-CoA and S-adenosylmethionine
*adrE*	Proq04g062850	1249	Ketoreductase	336	Formation of Andrastin F from Andrastin D
*adrF*	Proq04g062860a	952	Short-chain dehydrogenase	256	Formation of Andrastin D from Andrastin E
*adrG*	Proq04g062870	951	Prenyltransferase	316	Farnesylation of 3,5-dimethylorsellinic acid
*adrH*	Proq04g062880a	1633	FAD-dependent oxidoreductase	476	Formation of epoxyfarnesyl-3,5-dimethylorsellinic acid methyl ester
*adrI*	Proq04g062890	790	Terpene cyclase	245	Formation of Andrastin E epoxyfarnesyl-3,5-dimethylorsellinic acid methyl ester
*adrJ*	Proq04g062900	1554	Acetyltransferase	496	Formation of Andrastin C from Andrastin F
*adrK*	Proq04g062910	1050	Methyltransferase	278	Methylation of farnesyl-3,5-dimethylorsellinic acid

**Table 6 jof-09-00459-t006:** Annullatins D and F biosynthetic gene cluster in *P. roqueforti*.

Gene	ORF Name	Size (bp)	Protein	Size (aa)	Proposed Function
*anuK*	Proq03g054140	1246	Transcription factor (77.4% identity with FogI)	261	Unknown (transcriptional activation of the *anu* genes)
*anuA*	Proq03g054150	6704	Reducing polyketide synthase	2130	Formation of 2-hydroxymethyl-3-pentylphenol from acetyl-CoA and malonyl-CoA
*anuB*	Proq03g054160a	1458	Short-chain dehydrogenase/reductase	283	Formation of 2-hydroxymethyl-3-pentylphenol
*anuC*	Proq03g054170	276	Protein with 44.3% identity with the short-chain dehydrogenase/reductase FogB	91	Unknown (Formation of 2-hydroxymethyl-3-pentylphenol)
*anuD*	Proq03g054180	1288	Short-chain dehydrogenase	381	Unknown
*anuE*	Proq03g054190	1772	Cytochrome P450 hydroxylase	490	Formation annullatin E from 2-hydroxymethyl-3-pentylphenol into
*anuF*	Proq03g054200	937	Short-chain dehydrogenase/reductase	276	Formation of annullatin F
*anuG*	Proq03g054210	1797	Berberine bridge enzyme-like protein	509	Formation of annullatin D
*anuH*	Proq03g054220	1512	Prenyltransferase	451	Addition of a dimethylallyl group to annullatin E (formation of annullatin J) and formation of other prenylated derivatives
*anuI*	Proq03g054230	759	Short-chain dehydrogenase/reductase	252	Unknown
*anuJ*	Proq03g054240	2587	Aromatic ring hydroxylating dehydrogenase	818	Unknown

## Data Availability

Not applicable.

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
