# Peer review of "Secondary Metabolites Produced by the Blue-Cheese Ripening Mold Penicillium roqueforti; Biosynthesis and Regulation Mechanisms"

_jof, 2023, doi:10.3390/jof9040459_

Round 1
Reviewer 1 Report
The review "Secondary metabolites produced by the blue-cheese ripening mold Penicillium roqueforti; biosynthesis and regulation mechanisms", submitted to JoF by Chavez et al, is a complete compendium of the metabolites produced by a fungus of considerable applied interest because its participation in the production of certain so called blue cheeses. The description at the level of responsible gene clusters and the available information on metabolic pathways is very detailed, but in contrast the information on their regulation is limited. The review is reasonably comprehensive and may be worth of publication in JOF. However, I find different aspects that should be revised before publication:
1. The review is unnecessarily long due to the excessive detail in section 4, which describes biosynthetic gene clusters and metabolic pathways in a way cumbersome to read. The information is very repetitive with the figures, the text really reproduces what is already seen in the figures. Therefore, these long paragraphs should be simplified as much as possible. One possibility is to include a table with the genes, names, sizes, and functions of each cluster, in order to allow more succinct description in the texts. Additionally, irrelevant information is provided (for example, the intron numbers are of no interest in the context of the review, and are dispensable information.
Some specific observations in this section:
Strictly speaking, bidirectional promoters don't exist, they really are two different promoters, acting divergently. What is really interesting is that in these cases, the divergent promoters can share common regulatory elements.
Pseudogenes are not "residual", the idea of residual is already implicit in the word pseudogene. Even the term putative seems unnecesary if they include internal stop codons.
Pseudogenes also have direction, and should have their arrowheads in the figures.
It is striking that the genes in the clusters are described in such a way that it seems that there is a first gene and a last gene, when in fact the clusters have no beginning or end, or no right or left. It should start by saying something like, starting from gene XX, we find...
Lines 368: why may this ORF be a pseudogene?
In the cluster figures, different colors are used to indicate different genes using different criteria without an apparent logic. It would be more coherent to associate a certain color with a certain function with the same criterion in all figures. For example, identify with different colors the genes encoding (a) enzymes, (b) permeases, (c) regulatory proteins, (d) currently unidentified functions, and (e) pseudogenes (for this, black was already used). It would be helpful in the interpretation of the clusters.
Lines 527-528 The sentence "the full cluster was cloned under the control of the gpdA promoter and transferred to A. nidulans for heterologous expression" is evidently wrong. A cluster cannot be under the control of a single promoter. If anything, it would be the first gene alone that would be under the control of that promoter, which also makes little sense as an experimental strategy. The original article should be revised, and it should be confirmed that the first gene does not conserve its own promoter.
Other general comments:
It is a pity that some general regulators widely studied in fungi have not been analyzed in P. roquefortii. If this is not the case, the information should be extended to other Penicillium species if anything is available. There are no Penicillium studies that include any of these regulatory systems? In addition, the general regulators of nitrogen use, such as AreA, are not cited, and nitrogen regulation is very frequent in secondary metabolism.
Potential readers of this review may be particularly interested in the relationship between the secondary metabolites produced by R. roquefortii and the organoleptic properties of the cheeses in which they grow. No information is provided in this regard. If none is available, at least a brief mention should be made, possibly in the last section of "future challenges".
Line 562: there are no RNA-seq data deposited in the GEO data bank on Penicillium roquefortii or related Penicillium species that could provide information on the expression of secondary metabolism genes? I suppose that at some point in the future some work will emerge that provides information on the Penicillium transcriptome growing on cheese as a substrate.
Lines 767-768: Is it really credible that secondary metabolism has been studied in 7% of the existing fungal species? I do not think it is realistic.
Author Response
Response to Reviewer 1 Comments
The review "Secondary metabolites produced by the blue-cheese ripening mold Penicillium roqueforti; biosynthesis and regulation mechanisms", submitted to JoF by Chavez et al, is a complete compendium of the metabolites produced by a fungus of considerable applied interest because its participation in the production of certain so called blue cheeses. The description at the level of responsible gene clusters and the available information on metabolic pathways is very detailed, but in contrast the information on their regulation is limited. The review is reasonably comprehensive and may be worth of publication in JOF. However, I find different aspects that should be revised before publication:
Point 1. The review is unnecessarily long due to the excessive detail in section 4, which describes biosynthetic gene clusters and metabolic pathways in a way cumbersome to read. The information is very repetitive with the figures, the text really reproduces what is already seen in the figures. Therefore, these long paragraphs should be simplified as much as possible. One possibility is to include a table with the genes, names, sizes, and functions of each cluster, in order to allow more succinct description in the texts. Additionally, irrelevant information is provided (for example, the intron numbers are of no interest in the context of the review, and are dispensable information.
Response to 1: Thank you for the comment and suggestions. We have reorganized section 4 with tables including the information of each gene in the respective clusters.
Some specific observations in this section:
Point 2. Strictly speaking, bidirectional promoters don't exist, they really are two different promoters, acting divergently. What is really interesting is that in these cases, the divergent promoters can share common regulatory elements.
Response to 2: Thank you for the comment. We have revised the text and replaced the term bidirectional promoter with divergent promoter.
Point 3. Pseudogenes are not "residual", the idea of residual is already implicit in the word pseudogene. Even the term putative seems unnecesary if they include internal stop codons.
Response to 3: Thank you for the comment. We have removed the terms residual and putative.
Point 4. Pseudogenes also have direction, and should have their arrowheads in the figures.
Response to 4: Thank you for the comment. We have included the arrowheads in the pseudogenes in figures 2 and 6.
Point 5. It is striking that the genes in the clusters are described in such a way that it seems that there is a first gene and a last gene, when in fact the clusters have no beginning or end, or no right or left. It should start by saying something like, starting from gene XX, we find...
Response to 5: Thank you for the comment. We have proceed as suggested by the reviewer.
Point 6. Lines 368: why may this ORF be a pseudogene?
Response to 6: There is no reason to indicate that this gene may be a pseudogene. Therefore, this sentence has been removed from the text and from the legend to figure 5.
Point 7. In the cluster figures, different colors are used to indicate different genes using different criteria without an apparent logic. It would be more coherent to associate a certain color with a certain function with the same criterion in all figures. For example, identify with different colors the genes encoding (a) enzymes, (b) permeases, (c) regulatory proteins, (d) currently unidentified functions, and (e) pseudogenes (for this, black was already used). It would be helpful in the interpretation of the clusters.
Response to 7: Thank you for the comment. During the preparation of the figures we thought about this possibility. However, we discarded it since not all clusters contained the same type of genes. In addition, for the PR-gene cluster, we decided to use colors to identify genes according to who described it rather than by function.
Point 8. Lines 527-528 The sentence "the full cluster was cloned under the control of the gpdA promoter and transferred to A. nidulans for heterologous expression" is evidently wrong. A cluster cannot be under the control of a single promoter. If anything, it would be the first gene alone that would be under the control of that promoter, which also makes little sense as an experimental strategy. The original article should be revised, and it should be confirmed that the first gene does not conserve its own promoter.
Response to 8: Thank you for the comment. We agree with this reviewer. We have modified the text accordingly. In fact, the cluster was subcloned into plasmid pJN017, which contains the constitutive promoter of the gpdA gene. The cluster was subcloned without the natural promoter of anuK and since the first gene of the biosynthetic gene cluster corresponds to anuK, this gene was under the control of the gpdA promoter. This information is included in section 5.1.
Other general comments:
Point 9. It is a pity that some general regulators widely studied in fungi have not been analyzed in P. roquefortii. If this is not the case, the information should be extended to other Penicillium species if anything is available. There are no Penicillium studies that include any of these regulatory systems? In addition, the general regulators of nitrogen use, such as AreA, are not cited, and nitrogen regulation is very frequent in secondary metabolism.
Response to 9: Thank you for the comment. In general, there are no reports in P. roqueforti about general regulators widely studied in other fungi. We have added a paragraph about some of these regulators to section 5.3
“CreA, PacC, LaeA, or AreA have not yet been analyzed in this fungus, unlike in other fungi from the genus Penicillium, where their role in the control of secondary metabolism has been demonstrated. For example, CreA exerts carbon catabolic repression on penicillin biosynthesis and the expression of the pcbAB (the gene encoding the first enzyme of the penicillin pathway) in P. chrysogenum (Cepeda-García et al. 2014), while PacC exerts pH-dependent control on the production of patulin and the expression of its BGC in P. expansum (Chen et al. 2018). In the case of LaeA, its influence on the secondary metabolism of several Penicillium species has been documented [82]. Concerning the regulator of the nitrogen metabolism AreA, it has been involved in the control of the production of secondary metabolites in P. chrysogenum and P. griseofulvum [82]. Interestingly, the areA gene has been studied in P. roqueforti (Gente et al. 1999) but its role on secondary metabolism has not been addressed in this fungus.“
Point 10. Potential readers of this review may be particularly interested in the relationship between the secondary metabolites produced by R. roquefortii and the organoleptic properties of the cheeses in which they grow. No information is provided in this regard. If none is available, at least a brief mention should be made, possibly in the last section of "future challenges".
Response to 10: Thank you for the comment. We have added to Section 6. Future challenges and perspectives in the study of secondary metabolism in P. roqueforti
“P. roqueforti provides cheese with characteristic organoleptic properties. These properties have been associated to the potent proteolytic and lipolytic activities of this fungus [20, 21, Chávez et al. 2011]. In the case of the secondary metabolites produced by P. roqueforti, it remains to be determined if they have any role in the organoleptic properties of the cheeses. It is known that P. roqueforti inoculates secondary metabolites to cheese in which it grows [42, Fontaine et al. 2015] and therefore, one of the challenges to be addressed in this fungus is whether these secondary metabolites conferee organoleptic properties.”
Point 11. Line 562: there are no RNA-seq data deposited in the GEO data bank on Penicillium roquefortii or related Penicillium species that could provide information on the expression of secondary metabolism genes? I suppose that at some point in the future some work will emerge that provides information on the Penicillium transcriptome growing on cheese as a substrate.
Response to 11: To the best of our knowledge, there are no transcriptomics data deposited in the GEO data bank about the expression of secondary metabolism genes when P. roqueforti or related species are grown on cheese as a substrate. We have added a sentence in Section 6. Future challenges and perspectives in the study of secondary metabolism in P. roqueforti addressing this.
“It would be also interesting in the future to study the expression of those biosynthetic gene clusters when P. roqueforti or related species are grown on cheese as a substrate”.
Point 12. Lines 767-768: Is it really credible that secondary metabolism has been studied in 7% of the existing fungal species? I do not think it is realistic.
Response to 12: Thank you for the comment. This data come from literature and it has been estimated from data in 20214. Some authors have also found it controversial. Therefore, we have removed the reference and modified the sentence as follows:
“It has been estimated that the number of fungal species in nature is between 2 and 11 million species, of which around 150,000 are formally described taxa [Phukhamsakda et al. 2022]. Therefore, most of the biosynthetic potential of fungi is yet to be discovered.”

Reviewer 2 Report
This review provides a timely and interesting summary of the state of knowledge of secondary metabolites from P. roqueforti, an important organism in blue cheeses. The review is detailed and the biosynthetic pathways are interesting. The text is easy to read, with generally good English grammar, and is accompanied by clear diagrams. The number of references seems appropriate.
I think this review will be useful, and should be published essentially unchanged. Some minor points:
"Designed" on lines 56 and 61 is confused with "designated"
Line 59: "cleared" means clarified
line 81 "sheep of cow" does this mean sheep or cow?
line 105 "during the 70s of the twentieth century..." this is very awkward: during the 1970s,.... is much cleaner
Author Response
Response to Reviewer 2 Comments
This review provides a timely and interesting summary of the state of knowledge of secondary metabolites from P. roqueforti, an important organism in blue cheeses. The review is detailed and the biosynthetic pathways are interesting. The text is easy to read, with generally good English grammar, and is accompanied by clear diagrams. The number of references seems appropriate.
I think this review will be useful, and should be published essentially unchanged. Some minor points:
Point 1. "Designed" on lines 56 and 61 is confused with "designated"
Response to 1: Thank you for the correction. These typos have been corrected.
Point 2. Line 59: "cleared" means clarified
Response to 2: Thank you for the correction. This typo has been corrected.
Point 3. line 81 "sheep of cow " does this mean sheep or cow?
Response to 3: Thank you for the correction. This means sheep or cow and has been corrected.
Point 4. line 105 "during the 70s of the twentieth century..." this is very awkward: during the 1970s,.... is much cleaner.
Response to 4: Thank you for the comment. This sentence has been modified as suggested by the reviewer.

Round 2
Reviewer 1 Report
Having read the revised version of this excellent review, I am very pleased with the responses the authors have given to all the proposed changes and the improvements they introduced in the text. The review now seems to me more complete and understandable, and in my opinion it is acceptable for publication. I thank the authors for the effort they have made to attend all the points proposed for improvement.